# Scale-up costs and societal benefits of psychological interventions for alcohol use and depressive disorders in India

Siddhesh Zadey [1,2,3] *

1 GEMINI Research Center, Duke University School of Medicine, Durham, NC, United States of America,
2 Association for Socially Applicable Research (ASAR), Pune, Maharashtra, India, 3 Dr. D.Y. Patil Medical College, Hospital, and Research Centre, Pune, Maharashtra, India

* siddhesh.zadey@duke.edu

## Abstract

There is growing evidence for cost-effective psychological interventions by lay health workers for managing mental health problems. In India, Counseling for Alcohol Problems (CAP) and Healthy Activity Program (HAP) have been shown to have sustained cost-effectiveness for improving harmful alcohol use among males and depression remission among both sexes, respectively. We conducted a retrospective analysis of annual costs and economic benefits of CAP and HAP national scale-up with 2019 as the baseline. The CAP and HAP per capita integration costs were obtained from original studies, prevalence and disability-adjusted life-years for alcohol use disorders (AUD) and depressive disorders for 20–64 years old males and females from Global Burden of Disease study, and treatment gaps from National Mental Health Survey. We calculated three outcomes: 1) Programmatic scale-up costs for covering total or unmet needs. 2) Societal benefits from averted disease burden using human capital and value of life-year approaches. 3) Combinations of net benefits as differences between societal benefits and scale-up costs. Values were transformed to 2019 international dollars. CAP scale-up costs ranged from Int\$ 2.03 (95%UI: 1.67, 2.44) billion to Int\$ 6.34 (5.21, 7.61) billion while HAP ones ranged from Int\$ 6.85 (5.61, 8.12) billion to Int\$ 23.21 (19.03, 27.52) billion. Societal benefits due to averted AUD burden ranged from Int\$ 11.51 (8.75, 14.90) billion to Int\$ 38.73 (29.43, 50.11) billion and those due to averted depression burden ranged from Int\$ 30.89 (20.77, 43.32) billion to Int\$ 105.27 (70.78, 147.61) billion. All scenarios showed net positive benefits for CAP (Int\$ 6.05–36.38 billion) and HAP (Int\$ 11.12–93.50 billion) scale-up. The novel national-level scale-up estimates have actionable implications for mental health financing in India.

## 1 Introduction

India, the most populous country in the world, accounts for about 18% and 15% of the global burden of depressive and alcohol use disorders, respectively [1]. Yet, mental health expenditure only makes up 0.7% of the government health expenditure [2]. Further, the country with its 1.4 billion people, has only 7.5 psychiatrists per million people, much below the desired

**Data Availability Statement:** All data associated with this manuscript is present in the manuscript and the supplementary information (S1 Data).

**Competing interests:** I have read the journal's policy and the authors of this manuscript have the following competing interests: I represent the Association for Socially Applicable Research (ASAR) on the drafting committee of the Maharashtra State Mental Health Policy.

threshold of 30 psychiatrists per million people [3]. Hence, there has been growing interest in task-shifting interventions for the delivery of psychological treatments that can be scaled up in low-resource settings [4–7]. The past decade has seen cumulative evidence supporting the effectiveness and cost-effectiveness of contextually relevant psychological treatments for mental health and substance use problems in several countries, including India [8, 9].

Previous research has shown that brief psychological intervention for depression delivered by primary care lay counselors in the Healthy Activity Programme (HAP) was effective compared to enhanced usual care [9]. HAP was cost-effective at 12 months follow-up for lowering depression severity and increasing remission rates in men and women in the 18–65 age group with a probable diagnosis of moderately severe to severe depression measured by the Patient Health Questionnaire (PHQ-9) [10]. Similarly, brief intervention by lay counselors using motivational interviewing called Counseling for Alcohol Problems (CAP) was effective compared to enhanced usual care [8]. CAP was cost-effective at 12 months follow-up for increasing remission and abstinence rates in men in the 18–65 age group with harmful drinking determined by Alcohol Use Disorders Identification Test (AUDIT) [11]. Previous studies have demonstrated the effectiveness of well-monitored community-based mental health interventions delivered by professionally-trained lay counselors in low- and middle-income [12]. Hence, CAP and HAP can mitigate the treatment gaps, relieve the workload pressure on the existing limited psychiatric workforce, and lead to better population health and broader societal outcomes.

Beyond the evidence for effectiveness and cost-effectiveness, the scale-up would require an assessment of gross costs for deciding the feasibility. Additionally, the benefits provided by the scale-up will also have to justify costs. To that end, here, we present the preliminary estimates for scale-up costs of CAP and HAP at the national level. Additionally, we also investigate how these costs compare with the economic or societal benefits of burden averted due to treated depression and alcohol use problems in the Indian population, i.e., does the scale-up provide a net societal benefit? Through a range of sensitivity and uncertainty analyses, we produce a library of estimates for national-level programmatic scale-up costs, economic benefits due to disability-adjusted life-years (DALYs) averted, and net benefits.

## 2 Materials and methods

### 2.1 Variables and data sources

Mean per capita values of two kinds of costs were obtained from the studies that reported sustained (12 months follow-up) effectiveness and cost-effectiveness of CAP and HAP [10, 11]. First, the costs to the health system (HS) for integrating the program including consultations by doctors at primary health centers and hospitals, medicines, diagnostic tests, visits, admissions, and any other health service utilization components were obtained. Second, in addition to the HS costs, the time and transport costs, wages, and prolonged productivity losses to the patients and their family members were taken as societal costs. Next, we obtained the estimates for per capita gross domestic product (GDP), total health expenditure (THE), government health expenditure (GHE), and projected population values for 2019 for India curated by the Institute for Health Metrics and Evaluation (IHME) from the Global Health Data Exchange [13–15]. Mean and 95% uncertainty interval (2.5 and 97.5 percentile) estimates for prevalence and disability-adjusted life-years (DALYs) for depressive disorders and alcohol use disorders (AUD) in the 20–64 age group in 2019 were obtained, respectively, for both sexes and males in India from the Global Burden of Disease (GBD) study through the GBD Results Tool [1]. Population-level national values of self-reported treatment gaps for major depressive disorder (85.2%) and AUD (86.3%) were obtained from the National Mental Health Survey (2016) [16].

Values related to the value of life-year approach (see ahead in section 2.3) were based on the Lancet Commission on Investing in Health [17].

## 2.2 Analysis of scale-up costs

We calculated the annual (per year) scale-up costs of HAP and CAP for 2019 under two scale-up scenarios. In the first scenario, we defined the total need for HAP as the prevalence of depressive disorders and for CAP as the AUD prevalence. In the second scenario, we defined unmet need as the product of respective treatment gaps and disorder prevalence values. For both programs under both scenarios, we calculated two kinds of programmatic scale-up costs based on per capita HS and societal costs. The scale-up costs were calculated as the products of mean per capita costs and the (total or unmet) needs. A detailed outline of the analysis plan is presented in **Fig 1**. All costs were adjusted to 2019 international dollars ($Int). Uncertainty was propagated based on the 95% uncertainty interval values of prevalence estimates.

Scale-up costs were also assessed as percentages of aggregate GDP, THE, and GHE values. Aggregates were calculated as the products of mean per capita values and projected population (mean) counts for 2019.

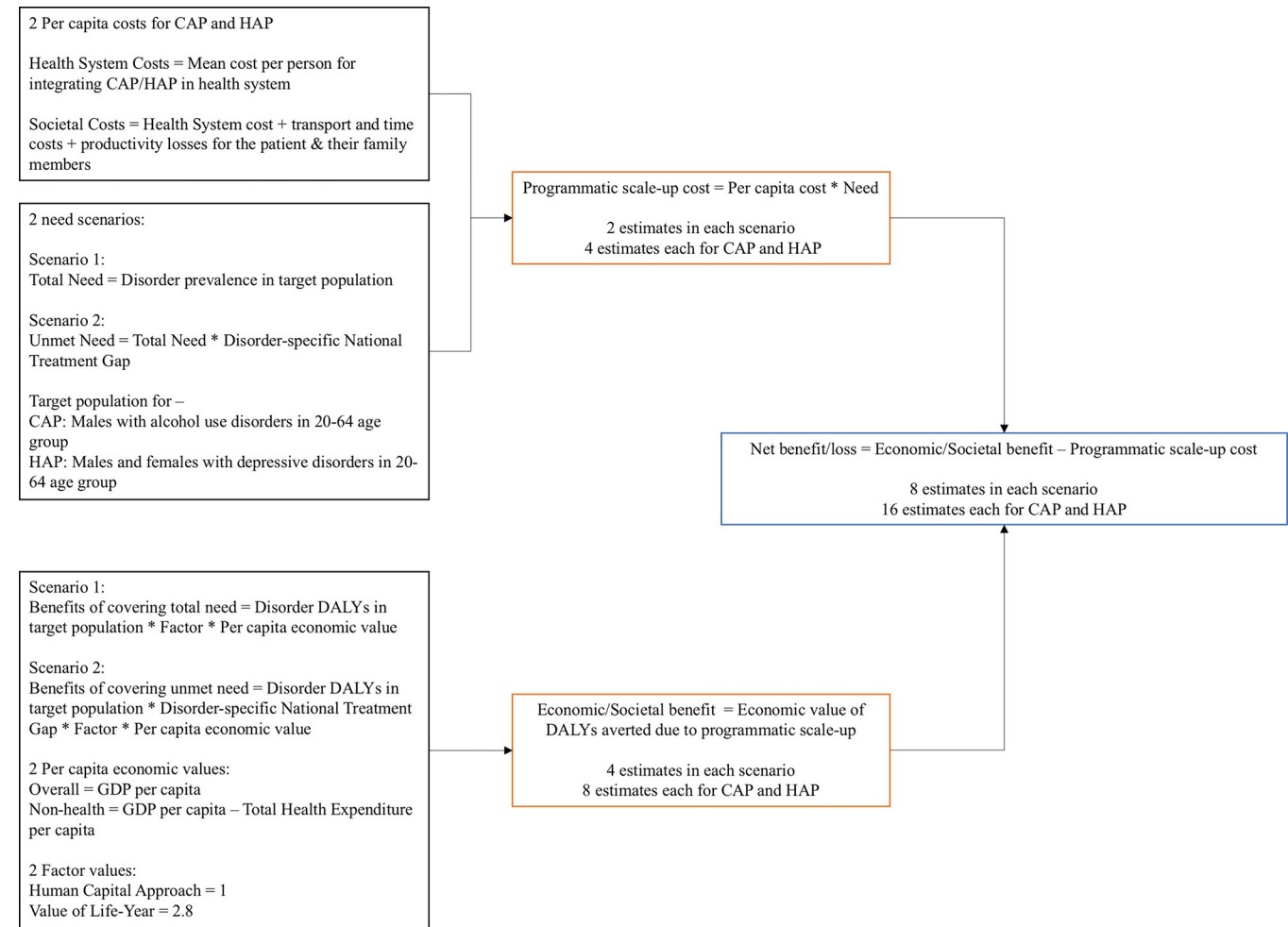

**Fig 1. Analysis plan.** Abbreviations: HS = Health System, CAP = Counseling for Alcohol Problems, HAP = Healthy Activity Program, GDP = Gross Domestic Product, THE = Total Health Expenditure, DALYs = Disability-Adjusted Life-Years.

## 2.3 Analysis of economic benefits of averted disease burden

The economic or societal benefits of scale-up of HAP and CAP were assessed by the benefits accrued due to aversion of depressive disorders and AUD disease burdens measured by DALYs, respectively. Economic or societal benefits were estimated using two related macroeconomic approaches: the human capital approach (HCA) and the value of life-year (VLY) or full-income approach [17–19]. HCA is based on the notion that a disability-adjusted life-year averted translates to economic productivity, particularly for the population groups that can contribute to or participate in the economy, i.e., the workforce. Previously, studies have used somewhat differing assumptions to define workforce populations [20, 21]. Parsimoniously, we assume that people in the 20–64 age group can participate in the Indian economy and contribute to the country's GDP. We calculated two kinds of economic/societal benefits. First, economic benefits were calculated as the product of averted disease burden in the 20–64 age group, measured as DALYs, and the non-health GDP per capita, i.e., the difference between per capita GDP and the per capita total health expenditure as considered previously in studies investigating the economic burden of suicide and COVID-19 [22, 23]. The second calculation used GDP per capita to account for more comprehensive benefit estimates similar to other studies investigating the economic burden of injuries [24].

The value of life-year or the full-income approach extends to include those who do not actively and currently participate in the economy [17, 18]. In other words, VLY puts a monetary value to 'any' life year beyond workforce/economic productivity years. Based on the enhancements in life expectancy achieved in the past decades, the Lancet Commission on Investing in Health estimated that for countries in the South Asian region, one life-year can be valued at about 2.8 times the GDP per capita calculated at a 3% discounting rate [17]. Hence, for the sake of clarity of calculations, HCA calculations were taken as (non-health and overall) GDP per capita multiplied by a factor of 1 and the VLY calculations as (non-health and overall) GDP per capita multiplied by a factor of 2.8. A detailed outline of the analysis plan is presented in **Fig 1**.

Economic or societal benefit calculations using the two GDP assumptions, under both HCA and VLY approaches, were performed for two scenarios—all (total need) DALYs averted and unmet need DALYs averted. In the first scenario, all DALYs averted were considered while the second scenario considered the DALYs averted assuming the programs scaled up to cover only the existing treatment gaps, i.e., product of treatment gap proportion, DALYs averted and (non-health or overall) GDP per capita (with a factor of 1 or 2.8). All benefit values were adjusted to 2019 Int$. We did not consider discounting for future years since our main goal was to look at annual estimates based on 2019 data. Uncertainty was propagated based on the 95% uncertainty interval values of DALY estimates.

## 2.5 Analysis of net benefits

We assessed the annual net benefits of scale-up by deducting the programmatic scale-up costs from the respective economic benefits estimates. For each program, i.e., CAP for AUD and HAP for depressive disorders, four economic or societal benefit values and two scale-up values under two scenarios (total need and unmet need) resulted in sixteen net benefit values (**Fig 1**). The net benefit values are presented in 2019 Int$.

Google Sheets (https://www.google.com/sheets/) was used for calculations and creating tables. Figures were created using Datawrapper (https://www.datawrapper.de/). The data used for and produced in this manuscript are presented in **S1 Data**.

**Fig 2. Annual scale-up costs in billion Int$ (2019) for CAP and HAP for covering total and unmet needs based on different cost considerations.** The solid bars depict mean values while the shaded regions depict a 95% uncertainty interval. Abbreviations: HS = Health System, CAP = Counseling for Alcohol Problems, HAP = Healthy Activity Program.

## 3 Results

### 3.1 Costs

The annual costs for the national scale-up of CAP and HAP to meet total and unmet needs are depicted in **Fig 2**. They ranged from Int$ 2.03 (95% Uncertainty Interval: 1.67, 2.44) billion for scaling up CAP to cover unmet need using HS costs to Int$ 23.21 (95% UI: 19.03, 27.52) billion for scaling up HAP to cover total need based on societal costs.

**Table 1** summarizes these costs in proportion to the 2019 gross domestic product, total health expenditure, and government health expenditure. Scale-up costs for CAP and HAP under multiple scenarios were less than 0.5% of the Indian GDP. However, these costs went up to 7% of the total and 25% of the government health expenditures.

**Table 1. National scale-up costs of CAP and HAP relative to India's gross domestic product (GDP), total health expenditure (THE), and government health expenditure (GHE).** Abbreviations: HS = Health System, CAP = Counseling for Alcohol Problems, HAP = Healthy Activity Program.

| Program | Scale-up scenario | % GDP | % THE | % GHE |
|---------|-------------------|-------|-------|-------|
| CAP | HS Costs—Total Need | 0.02 | 0.68 | 2.46 |
| | Societal Costs—Total Need | 0.07 | 1.83 | 6.63 |
| | HS Costs—Unmet Need | 0.02 | 0.59 | 2.12 |
| | Societal Costs—Unmet Need | 0.06 | 1.58 | 5.72 |
| HAP | HS Costs—Total Need | 0.08 | 2.32 | 8.40 |
| | Societal Costs—Total Need | 0.24 | 6.71 | 24.27 |
| | HS Costs—Unmet Need | 0.07 | 1.98 | 7.16 |
| | Societal Costs—Unmet Need | 0.20 | 5.72 | 20.68 |

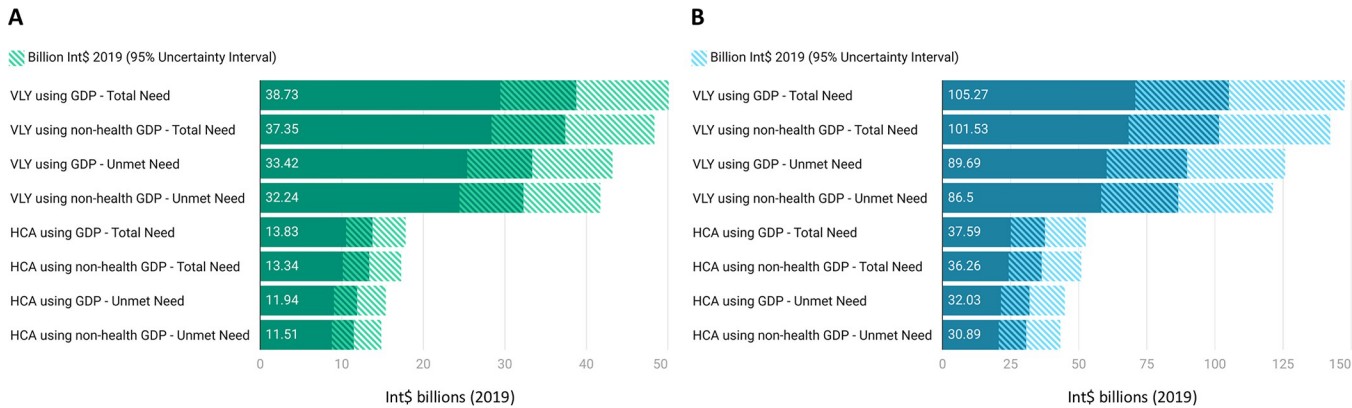

**Fig 3. Economic or societal benefits (in billion Int$ 2019) due to the averted burden of A) alcohol use disorders in males by scaling up CAP and B) depressive disorders in males and females by scaling up HAP under different considerations.** The solid bars depict mean values while the shaded regions depict a 95% uncertainty interval. GDP refers to GDP per capita here. Abbreviations: HCA = Human Capital Approach, VLY = Value of Life-Year, CAP = Counseling for Alcohol Problems, HAP = Healthy Activity Program, GDP = Gross Domestic Product.

## 3.2 Economic benefits of averted disease burden

The societal or economic benefits from averting the burden of AUD in males and depressive disorder in males and females through CAP and HAP scale-up under different approaches are presented in **Fig 3A and 3B**, respectively. For AUD in 2019, the benefits ranged from Int$ 11.51 (95% UI: 8.75, 14.90) billion by covering unmet need under the HCA framework using non-health GDP to Int$ 38.73 (95% UI: 29.43, 50.11) billion by covering total need under the VLY approach using overall GDP per capita (**Fig 3A**). For depressive disorders, the benefits ranged from Int$ 30.89 (95% UI: 20.77, 43.32) billion by covering unmet need under the HCA framework using non-health GDP to Int$ 105.27 (95% UI: 70.78, 147.61) billion by covering total need under the VLY approach using overall GDP per capita (**Fig 3B**).

## 3.3 Net benefits

Annual net benefits based on the differences between the mean values of all possible economic or societal benefits and corresponding scale-up costs are given in **Fig 4**. Under all scenarios, for both CAP and HAP scale-up, there was a net positive benefit. The smallest net benefit of Int$ 6.05 billion was observed when economic benefits due to averted AUD burden were estimated by HCA using non-health GDP per capita and CAP scale-up was considered to cover unmet need using societal costs. The largest net benefit of Int$ 97.23 billion was observed when economic benefits due to averted depressive disorder burden were estimated by VLY using GDP per capita and HAP scale-up was considered to cover total need using HS costs.

## 4 Discussion

### 4.1 Summary and interpretation

In this preliminary analysis, we estimated the annual costs of scaling up CAP and HAP at the national level and showed that these costs fall much below the limits of total health expenditure and India's GDP. Under multiple scenarios including total and unmet need, different approaches including the restrictive HCA and the broader VLY, the scale-up costs were smaller than the societal or economic benefits due to averted disease burden for both AUD and depressive disorders, resulting in net societal benefits. It is important to know that these calculations do not follow the approach of a stochastic model investigating the: a) accumulating benefits from treatment extended beyond the year, b) possible relapse after the year, c)

**Billion Int$ 2019**

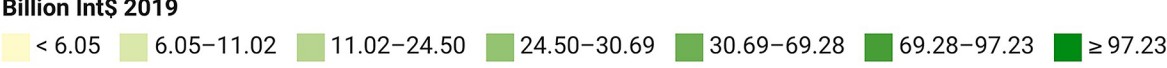

| Benefit-Cost Combinations | CAP Net Benefits - Total Need | CAP Net Benefits - Unmet Need | HAP Net Benefits - Total Need | HAP Net Benefits - Unmet Need |
|---|---|---|---|---|
| Benefits by HCA using GDP - Scale-up using HS Costs | 11 | 10 | 30 | 25 |
| Benefits by HCA using non-health GDP - Scale-up using HS Costs | 11 | 9 | 28 | 24 |
| Benefits by VLY using GDP - Scale-up using HS Costs | 36 | 31 | 97 | 83 |
| Benefits by VLY using non-health GDP - Scale-up using HS Costs | 35 | 30 | 93 | 80 |
| Benefits by HCA using GDP - Scale-up using Societal Costs | 7 | 6 | 14 | 12 |
| Benefits by HCA using non-health GDP - Scale-up using Societal Costs | 7 | 6 | 13 | 11 |
| Benefits by VLY using GDP - Scale-up using Societal Costs | 32 | 28 | 82 | 70 |
| Benefits by VLY using non-health GDP - Scale-up using Societal Costs | 31 | 27 | 78 | 67 |

**Fig 4. Net benefits (in billion Int$ 2019) of CAP and HAP scale-up for averting alcohol use disorders and depressive disorders burden across multiple benefit-cost combinations.** Mean values of scale-up costs and economic benefits used. GDP refers to GDP per capita here. Abbreviations: HCA = Human Capital Approach, VLY = Value of Life-Year, CAP = Counseling for Alcohol Problems, HAP = Healthy Activity Program, GDP = Gross Domestic Product, HS = Health System.

limited economic productivity or labor force participation post-remission, or d) lag in joining the economy after successful treatment. Rather, our argument here is that these psychological interventions could help cover the treatment gap that would, in turn, avert the existing burden of disease.

## 4.2 Context and implications

There has been growing evidence-based for investing in mental healthcare by integrating alternative models of care such as lay health worker-led psychosocial interventions beyond the

traditional psychiatric consultations to reduce access disparities and improve treatment gaps [6, 7]. This evidence has also contributed to changing mental health policies and upgrading laws [5, 25]. In India, the National Mental Health Policy (2014) and the Mental Healthcare Act (2017) recognize the country's mental health needs and adopt a biopsychosocial approach embedded in a rights-based framework to ensure universal care [26, 27]. However, the Policy and the Act have provided limited inputs for financing, targeted interventions, and contextualization as per local needs [28, 29]. Previously, it has been estimated that implementing the Mental Healthcare Act would require Rs. 94000 crores or US$11.34 billion (1 Indian rupee = 0.012 US$) with 6.5 times returns on investment [30]. Overall, government health expenditure (GHE) in India is 1.28% of the GDP which is much lower than several other lower-middle-income countries and some low-income countries [31]. The expenditure constricts further in the case of mental health. For instance, the current government expenditure on mental health in the 2022–23 budget was Rs. 1035.39 crore, which forms 0.7% of the Ministry of Health and Family Welfare budget [2]. In our findings, we demonstrated that CAP and HAP scale-up costs formed only small percentages of the GDP or THE, but the proportions went significantly up when compared to GHE, pointing to the limited government expenditure. Hence, increasing government expenditure on health and particularly mental health is urgently needed.

While there has been greater attention towards mental health issues during and after the COVID-19 pandemic, the gains in the mental healthcare budget have been modest, expansion of coverage is urgently needed, and quality of service delivery requires enhancement necessitating expansion of the mental healthcare budget [32, 33]. However, such an expansion also warrants appropriate allocation that matches the problem magnitude and ensures a return on investment. Depressive disorders contribute to over a third of the disease burden due to all mental disorders while alcohol use disorders make up more than half of all substance use disorders' DALYs [1]. Our analysis demonstrates the benefits of implementing specific psychological interventions for depressive and alcohol use disorders. Hence, scaling up interventions for these disorders should be prioritized.

For feasible scale-up, interventions like CAP and HAP can be integrated into existing programs. For instance, District Mental Health Programme run in multiple Indian states, can be strengthened by the integration of CAP and HAP to ensure a reduction in disparities in access to care [34]. Certain programs, such as the government-run Drug De-addiction Programme, can be expanded to reach beyond hospitals and treatment centers by integrating CAP into primary care [35]. Integration of such interventions in community-based mental health programs for other issues such as suicide prevention is also critical for providing early care to population groups at risk [36]. Joint implementation of the CAP and HAP programs can help reduce system-wide costs such as those associated with the training of lay health workers. Further, there is also upcoming evidence on the transdiagnostic acceptability and safety of these programs, e.g., the effect of CAP on comorbid depression in people with AUD, which can potentially add to the economic or societal benefits due to averted disease burden if it is found to be cost-effective [37].

### 4.3 Strengths and limitations

This study has some strengths. First, the study reports novel national-level scale-up costs for specific interventions which have actionable implications for mental health financing. Second, we provided a library of estimates along with uncertainty analyses for scale-up costs and economic benefits that ensure the robustness of findings and allow the decision-makers to choose from scenarios. Third, for scale-up costs, we used cost-effectiveness for 12-month follow-up,

i.e., sustained cost-effectiveness, which was more suitable for annual estimates than the 3-month follow-up cost-effectiveness reported in the primary trial findings.

The study also has several limitations. First, the cost-effectiveness data on CAP and HAP are based on specific inclusion criteria based on disease severity. For instance, CAP included people with a score in the 12–19 range on AUDIT while the HAP study included those with a PHQ-9 score greater than 14 [10, 11]. However, for the treatment gap, values were obtained from the national survey which only had the disorder classification (AUD and major depressive disorder) without an actual disease severity gradation [16]. Second, there was also a small mismatch between the age groups and gender vs. sex distinction. CAP included men while HAP included men and women belonging to the 18–65 years age group. However, the data on prevalence and DALYs from the GBD study were available for the 20–64 age group which had a small deviation from the original study participant profile. Further, the GBD data was presented for sexes (males and females) and not genders (men and women). Hence, though not accurate or completely appropriate, sex and gender categories were used interchangeably here due to a lack of data. Third, it was assumed for the sake of feasibility of calculation that all existing or unmet disease burdens can be averted by psychological interventions, which may not be true. Fourth, the analysis did not account for the mediation effect of readiness to change in the case of CAP and that of behavioral activation for HAP [10, 11]. In other words, for parsimony, it was assumed that all people with AUD had the same level of readiness to change while those with depression had similar behavioral activation. Fifth, the human capital approach did not include discounting for future years. However, the factor value chosen for the value of life-year approach had a 3% discount rate. Sixth, the willingness to pay thresholds underlying cost-effectiveness for CAP and HAP were based on just one Indian state–Goa [8, 9]. Ideally, national scale-up values should use nationally-representative cost-effectiveness values. Seventh, data on the treatment gap was from 2016 and does not temporally match other data (i.e., 2019). However, the National Mental Health Survey was the only most recent nationally-representative source providing such data. Eighth, due to the difficulty in computing the distributions, no uncertainty analysis was conducted for net benefits.

## 5 Conclusions

This study showed that the annual costs of scaling up lay health worker-delivered psychological interventions for depression and alcohol use disorders at the national level were lower than the societal or economic benefits due to averted disease burden. Therefore, the net benefits of scaling up are substantial. The costs also formed only a small portion of the total health expenses and the country's GDP. Further studies should expand on these findings using sophisticated modeling approaches and estimate state-wise scale-up costs for informing local health planning and financing. Such economic assessments are essential to improve mental health planning in low- and lower-middle-income countries.

## Supporting information

**S1 Data. Dataset used for the analysis.**
(CSV)

## Author Contributions

**Conceptualization:** Siddhesh Zadey.

**Formal analysis:** Siddhesh Zadey.

**Investigation:** Siddhesh Zadey.

**Methodology:** Siddhesh Zadey.

**Project administration:** Siddhesh Zadey.

**Supervision:** Siddhesh Zadey.

**Writing – original draft:** Siddhesh Zadey.

**Writing – review & editing:** Siddhesh Zadey.

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
