## [Decision Letter · Decision Letter 0]

14 Jun 2023

PGPH-D-23-00893

Scale-up costs and societal benefits of psychological interventions for alcohol use and depressive disorders in India

Dear Dr. zadey,

Thank you for submitting your manuscript to PLOS Global Public Health. After careful consideration, we feel that it has merit but does not fully meet PLOS Global Public Health’s publication criteria as it currently stands. Therefore, we invite you to submit a revised version of the manuscript that addresses the points raised during the review process. 

We look forward to receiving your revised manuscript.

Kind regards,

Ejemai Eboreime, MD, MSc, PhD

Academic Editor

Journal Requirements:

2. Please send a completed 'Competing Interests' statement, including any COIs declared by your co-authors. If you have no competing interests to declare, please state "The authors have declared that no competing interests exist". Otherwise please declare all competing interests beginning with the statement "I have read the journal's policy and the authors of this manuscript have the following competing interests:"

Additional Editor Comments (if provided):

Reviewers' comments:

Reviewer's Responses to Questions

**Comments to the Author**

1. Does this manuscript meet PLOS Global Public Health’s publication criteria? Is the manuscript technically sound, and do the data support the conclusions? The manuscript must describe methodologically and ethically rigorous research with conclusions that are appropriately drawn based on the data presented.

Reviewer #1: Yes

Reviewer #2: Partly

2. Has the statistical analysis been performed appropriately and rigorously?

Reviewer #1: Yes

Reviewer #2: No

3. Have the authors made all data underlying the findings in their manuscript fully available (please refer to the Data Availability Statement at the start of the manuscript PDF file)?

Reviewer #1: Yes

Reviewer #2: No

4. Is the manuscript presented in an intelligible fashion and written in standard English?

Reviewer #1: Yes

Reviewer #2: Yes

5. Review Comments to the Author

Reviewer #1: The manuscript was clear, well written according to the PLOS Global public health's publication criteria. Health expenditure and country's GDP and the annual costs of scaling up CAP and HAP were estimated.

Reviewer #2: ABSTRACT

The author discusses the necessity for CAP and HAP finance to be scaled up in India, as well as estimations of the total benefits of such financial scaling.

Lines 10 to 19 of the abstract must be updated to provide a clear comprehension of the desired methodology and outcomes. For example, ‘Scale-up costs were calculated for meeting total or unmet needs, societal benefit estimates based on averted disease burden were calculated using human capital and the value of life-year approaches’ doesn’t entirely make sense.

INTRODUCTION

1. Paragraph 1, line 3 asserts that India requires three psychiatrists per million, but I disagree. According to data from OECD countries and the USA, the ideal number is between 10-20 per 100,000. Additionally, the study cited in the paragraph estimates that there are only three psychiatrists per 100,000 in India. The paper also laments the scarcity of doctors, which makes it challenging to meet the current standards, which are generally inadequate. The sentence should be updated to reflect accurate facts.

2. The first line of the second paragraph is too long and difficult to understand. To improve clarity, authors should use shorter sentences and choose their words carefully. For instance:

Research has shown that primary care lay-counseling with a brief intervention called HAP is both effective and cost-effective, as compared to ABC.

3. In the same way, it is possible to rephrase line 7 in paragraph 2.

4. “Scaling up programs such as CAP and HAP at the national level can mitigate the treatment gaps, relieve the workload pressure on the existing limited psychiatric workforce, and lead to better population health and broader societal outcomes.”

I feel the above should be presented as monitored or governed, professionally trained lay-counselling services scaling up. See reference below.

Connolly, S. M., Vanchu-Orosco, M., Warner, J., Seidi, P. A., Edwards, J., Boath, E., & Irgens, A. C. (2021). Mental health interventions by lay counselors: a systematic review and meta-analysis. Bulletin of the World Health Organization, 99(8), 572–582. https://doi.org/10.2471/BLT.20.269050

MATERIALS AND METHODS

General comment

Mathematics or diagrammatic summary of the calculations would prove essential to follow the arguments raised as far as the scale-up estimates are concerned. Similarly, the same applies to the economic benefits.

I didn’t follow how all the scale up estimates were done based on 2016 data as referenced in the sentence “Population-level national values of self-reported treatment gaps for major depressive disorder (85.2%) and AUD (86.3%) were obtained from the National Mental Health Survey (2016) [15]” in section 2.1.

Additionally, for the sentence “Values related to the value of life-year approach (see ahead) were based on previous studies [16]”, please provide a brief description and what values were used in your study.

I would find it interesting to demonstrate (as earlier mentioned) how each variable used was estimated and grounds that support the use of the variables in literature.

6. PLOS authors have the option to publish the peer review history of their article (what does this mean?). If published, this will include your full peer review and any attached files.

**Do you want your identity to be public for this peer review?** For information about this choice, including consent withdrawal, please see our Privacy Policy.

Reviewer #1: No

Reviewer #2: **Yes: **Kemal Aydın

---

## [Decision Letter · Decision Letter 1]

24 Aug 2023

Scale-up costs and societal benefits of psychological interventions for alcohol use and depressive disorders in India

PGPH-D-23-00893R1

Dear Dr zadey,

We are pleased to inform you that your manuscript 'Scale-up costs and societal benefits of psychological interventions for alcohol use and depressive disorders in India' has been provisionally accepted for publication in PLOS Global Public Health.

Best regards,

Ejemai Eboreime, MD, MSc, PhD

Academic Editor

Reviewer Comments (if any, and for reference):

Reviewer's Responses to Questions

**Comments to the Author**

1. If the authors have adequately addressed your comments raised in a previous round of review and you feel that this manuscript is now acceptable for publication, you may indicate that here to bypass the “Comments to the Author” section, enter your conflict of interest statement in the “Confidential to Editor” section, and submit your "Accept" recommendation.

Reviewer #2: All comments have been addressed

2. Does this manuscript meet PLOS Global Public Health’s publication criteria? Is the manuscript technically sound, and do the data support the conclusions? The manuscript must describe methodologically and ethically rigorous research with conclusions that are appropriately drawn based on the data presented.

Reviewer #2: Yes

3. Has the statistical analysis been performed appropriately and rigorously?

Reviewer #2: Yes

4. Have the authors made all data underlying the findings in their manuscript fully available (please refer to the Data Availability Statement at the start of the manuscript PDF file)?

Reviewer #2: Yes

5. Is the manuscript presented in an intelligible fashion and written in standard English?

Reviewer #2: Yes

6. Review Comments to the Author

Reviewer #2: (No Response)

7. PLOS authors have the option to publish the peer review history of their article (what does this mean?). If published, this will include your full peer review and any attached files.

**Do you want your identity to be public for this peer review?** For information about this choice, including consent withdrawal, please see our Privacy Policy.

Reviewer #2: No
